# Characterization of *Cronobacter sakazakii* and *Cronobacter malonaticus* Strains Isolated from Powdered Dairy Products Intended for Consumption by Adults and Older Adults

**DOI:** 10.3390/microorganisms11122841

**Published:** 2023-11-23

**Authors:** Julio Parra-Flores, Fernanda Flores-Soto, Carolina Flores-Balboa, María P. Alarcón-Lavín, Adriana Cabal-Rosel, Beatriz Daza-Prieto, Burkhard Springer, Ariadnna Cruz-Córdova, José Leiva-Caro, Stephen Forsythe, Werner Ruppitsch

**Affiliations:** 1Department of Nutrition and Public Health, Universidad del Bío-Bío, Chillán 3800708, Chile; mpalarcon@ubiobio.cl; 2Nutrition and Dietetics School, Universidad del Bío-Bío, Chillán 3800708, Chile; fc.floressoto@gmail.com (F.F.-S.); floresbalboacarolina@gmail.com (C.F.-B.); 3Austrian Agency for Health and Food Safety, Institute for Medical Microbiology and Hygiene, 1220 Vienna, Austria; adriana.cabal-rosel@ages.at (A.C.-R.); beatriz.daza-prieto@ages.at (B.D.-P.); burkhard.springer@ages.at (B.S.); werner.ruppitsch@ages.at (W.R.); 4Intestinal Bacteriology Research Laboratory, Hospital Infantil de México Federico Gómez, Mexico City 06720, Mexico; ariadnnacruz@yahoo.com.mx; 5Department of Nursing, Universidad del Bío-Bío, Chillán 3800708, Chile; jleiva@ubiobio.cl; 6FoodMicrobe.com Ltd., Adams Hill, Keyworth, Nottingham NG12 5GY, UK

**Keywords:** *Cronobacter sakazakii*, *Cronobacter malonaticus*, MDR, virulence factors, powder dairy products, whole-genome sequencing, older adults

## Abstract

The objective of this study was to characterize *Cronobacter* spp. and related organisms isolated from powder dairy products intended for consumption by adults and older adults using whole-genome sequencing (WGS), and to identify genes and traits that encode antibiotic resistance and virulence. Virulence (VGs) and antibiotic resistance genes (ARGs) were detected with the Comprehensive Antibiotic Resistance Database (CARD) platform, ResFinder, and MOB-suite tools. Susceptibility testing was performed using disk diffusion. Five presumptive strains of *Cronobacter* spp. were identified by MALDI–TOF MS and ribosomal MLST. Three *C. sakazakii* strains were of the clinical pathovar ST1, one was ST31, and the remaining isolate was *C. malonaticus* ST60. In addition, *Franconibacter helveticus* ST345 was identified. The *C. sakazakii* ST1 strains were further distinguished using core genome MLST based on 2831 loci. Moreover, 100% of the strains were resistant to cefalotin, 75% to ampicillin, and 50% to amikacin. The *C. sakazakii* ST1 strains were multiresistant (MDR) to four antibiotics. Additionally, all the strains adhered to the N1E-115 cell line, and two invaded it. Eighteen ARGs mainly involved in antibiotic target alteration and antibiotic efflux were detected. Thirty VGs were detected and clustered as flagellar proteins, outer membrane proteins, chemotaxis, hemolysins, and genes involved in metabolism and stress. The pESA3, pSP291-1, and pCMA1 plasmids were detected, and the prevalent mobile genetic elements (MGEs) were ISEsa1, ISEc52, and IS26. The isolates of *C. sakazakii* and *C. malonaticus* exhibited multiresistance to antibiotics, harbored genes encoding various antibiotic resistance proteins, and various virulence factors. Consequently, these contaminated powdered dairy products pose a risk to the health of hypersensitive adults.

## 1. Introduction

*Cronobacter* spp. is a genus of enteropathogenic bacteria comprising seven species: *Cronobacter sakazakii*, *C. malonaticus*, *C. universalis*, *C. turicensis*, *C. muytjensii*, *C. dublinensis*, and *C. condimenti* [1,2,3]. Among these species, *C. sakazakii* and *C. malonaticus* are especially important in clinical settings [4].

*C. sakazakii* primarily affects premature newborns and infants, while *C. malonaticus* is more common in older adults [5,6,7]. In infants, meningitis, septicemia, and necrotizing enterocolitis (NEC) have been reported, while urinary tract and respiratory infections are more common in adults. The mortality rates associated with *C. sakazakii* infection in neonatal meningitis and septicemia are 15% and 25%, respectively [6]. Neonatal meningitis cases are especially linked to *C. sakazakii* sequence type 4 (ST4/CC4) infections [8], while cases of infections in adults are often associated with *C. malonaticus* ST7 and ST60 [9,10]. In 2014, Holy et al. reported the presence of *Cronobacter* spp. in all samples from hospitalized patients between 2005 and 2011. The prevalence was higher in the under-one-year-old group at 7.7%, with an incidence of 8.7 per 1000 patients. For adults, the incidence of *Cronobacter* spp. was 0.5, 2.8, and 2.0 in patients aged 45–64, 65–74, and over 75 years, respectively [11]. Even though infections by *C. sakazakii* have been associated with particularly sensitive groups, in 2016, an outbreak of gastroenteritis affecting 156 Chinese students caused by *Cronobacter sakazakii* and *C. malonaticus* was reported [12]. *Cronobacter* spp. infections in infants are primarily attributed to the consumption of contaminated rehydrated powdered infant formula (PIF). *Cronobacter* spp. can be isolated from powdered infant formula (PIF), rehydrated powdered milk, infant cereals, water, and various surfaces within manufacturing facilities. Furthermore, international studies on *Cronobacter* spp. have reported prevalence values ranging from 3% to 30% [13,14]. However, the probable source of infection in adults has not been determined. Furthermore, the ability of these pathogens to persist in food environments and settings poses an additional risk to the presence of this pathogen in food [15].

The severity of the clinical condition has been associated with the presence of plasmid-encoded virulence factors [16,17], adherence and invasion in cell lines [18,19], and various other genes, such as *aut*, *cpA*, *fliC*, *hly*, *ompA*, *sip*, *plas*, and *inv,* that are important for the adherence of the microorganism to the epithelial surface, multiplication, colonization, evasion of the host’s natural or innate defenses, tissue invasion, and cellular damage [20,21,22]. Other factors include the utilization of sialic acid as a carbon source, capsule composition and presence, and endotoxin production [23]. Another important aspect is the resistance to β-lactam antibiotics such as cephalothin, cefotaxime, ceftazidime, and ampicillin, along with the presence of resistance genes such as *marA*, *glpT*, *ampH*, *blaCSA*, *blaCMA*, *mrc*, and the efflux pump system *AcrAB-TolC* [24,25,26].

The use of whole-genome sequencing (WGS) has allowed for more precise differentiation between species. For *Cronobacter*, WGS has shown high discrimination of conserved and variable genetic information, enabling differentiation between closely related species such as *Franconibacter helveticus* and *Enterobacter hormaechei*. Therefore, WGS is used as a tool for identification, genotyping (seven-loci multilocus sequence typing (MLST) and core genome multilocus sequence typing), CRISPR-Cas array profiling, serogrouping, SNP determination, detection of genes conferring resistance to antibiotics (ARGs) and/or virulence (VGs), and therefore considerably facilitates molecular epidemiological studies [27].

In this study, the objective was to use WGS to characterize *Cronobacter* spp. strains isolated from powdered dairy products intended for consumption by adults and older adults, and to identify genes and traits that encode antibiotic resistance and virulence.

## 2. Materials and Methods

### 2.1. Sampling 

A total of 100 samples of dairy products intended for adults and older adults were analyzed between 2018 and 2020, from 2 commercially available brands. Each container or can served as a sample and was obtained monthly from supermarkets and pharmacies due to their monthly restocking practices. This allowed for greater variability in terms of the production batch origins of the samples. The samples were stored in their sealed containers following the manufacturer’s guidelines until analysis. Later, the containers were opened within a laminar flow hood. The tools used for sample collection were sterilized at 121 °C for 15 min, and the outer surfaces of the containers were treated with 70% alcohol.

### 2.2. Isolation and Identification Methods of Cronobacter *spp.*

*Cronobacter* spp. strains were isolated following the general methodology described by Iversen et al. [28]. Initially, food and environmental samples were pre-enriched in buffered peptone water (BPW) and then transferred to *Enterobacteriaceae* enrichment broth (BD Difco, Sparks, MD, USA). Subsequently, the samples were plated onto Brilliance CM 1035 chromogenic agar (Oxoid Thermo-Fisher, Basingstoke, UK) and purified on trypticase soy agar (BD Difco, Sparks, MD, USA). Before sequencing, the suspected strains underwent presumptive identification using Matrix-Assisted Laser Desorption/Ionization–Time of Flight Mass Spectrometry (MALDI–TOF MS) (Bruker, Billerica, MA, USA) and the MALDI Biotyper Compass IVD 4.1.60 software (Bruker, Billerica, MA, USA) as described by Lepuschitz et al. [29]. The confirmation of presumptive *Cronobacter* spp. strains was achieved using the ribosomal multilocus sequence typing (rMLST) software available at https://pubmlst.org/species-id (accessed on 28 September 2023) [30] and average nucleotide identity (ANI) analysis using JSpecies v3.9.7 [31]. The strains were annotated using software Proksee v1.3.0 [32] and Prokka v1.1.1 [33]. 

### 2.3. Whole-Genome Sequencing (WGS)

*Cronobacter* spp. isolates were cultured on Columbia blood agar plates (bioMérieux, Marcy-l’Étoile, France) at 37 °C for 24 h. Genomic DNA was extracted from bacterial cultures using the MagAttract HMW DNA Kit (Qiagen, Hilden, Germany) following the manufacturer’s instructions. The concentration of DNA was quantified using a Qubit 2.0 fluorometer (Thermo Fisher Scientific, Waltham, MA, USA) and the dsDNA BR assay kit (Thermo Fisher Scientific). For library preparation, Nextera XT chemistry (Illumina Inc., San Diego, CA, USA) was employed, and the libraries were subjected to a 2 × 300 bp paired-end sequencing run on an Illumina MiSeq sequencer. Sequencing was performed to achieve a minimum coverage of 80-fold using standard Illumina protocols. Raw reads were quality-controlled using FastQC v0.11.9, and adapter sequences were removed, while the last 10 bp of each sequence and sequences with quality scores below 20 were trimmed using Trimmomatic v0.36 [34]. The reads were then assembled using SPAdes v3.11.1 [35]. Finally, contigs were filtered based on a minimum coverage of 5× and a minimum length of 200 bp using SeqSphere+ software v9.0.8 (Ridom GmbH, Würzburg, Germany) [36]. Sequencing quality data as coverage, no. of the contigs, N50 were included Appendix A.

### 2.4. Sequence Type (ST) and Core Genome Multilocus Sequence Typing (cgMLST) of Cronobacter Isolates

Comprehensive core genome multilocus sequence typing (cgMLST) was carried out on *Cronobacter* isolates using a core genome comprising 2831 target genes using the Ridom SeqSphere+ software v.9.0.8 (Ridom, Münster, Germany) [36]. A minimum spanning tree (MST) was created from the allelic profiles of the isolates to determine genotypic relationships. Clusters were defined as isolates with a maximum difference of 10 alleles. Additionally, the sequences of the seven housekeeping genes commonly used in conventional MLST for *C. sakazakii* and *C. malonaticus* (*atpD*, *fusA*, *glnS*, *gltB*, *gyrB*, *infB*, and *ppsA*) were extracted and cross-checked against the *Cronobacter* MLST database available at https://pubmlst.org/organisms/cronobacter-spp/ (accessed on 1 September 2023).

### 2.5. Determination of Serotypes

The serotypes of *Cronobacter* spp. were determined by analyzing the profiles of the *gnd* and *galF* genes, which are specific to the serotype O region [37]. Whole-genome sequencing (WGS) data analysis was performed using the BIGSdb tool available in the PubMLST database (https://pubmlst.org/organisms/cronobacter-spp) (accessed on 10 September 2023) [38].

### 2.6. Antibiotic Susceptibility

Antibiotic susceptibility was assessed using the disk diffusion method, following the recommendations of the Clinical and Laboratory Standards Institute [39]. Commercial antibiotic disks include amikacin (30 µg), ampicillin (10 μg), amoxicillin-clavulanic acid (20/10 µg), ceftazidime (30 µg), ciprofloxacin (5 µg), chloramphenicol (30 μg), cefotaxime (30 μg), cefepime (30 µg), gentamicin (10 μg), cephalothin (30 μg), and tetracycline (30 µg). The interpretation of resistance/susceptibility profiles was conducted according to the manufacturer’s instructions. *Escherichia coli* ATCC 25922 and *Pseudomonas aeruginosa* ATCC 27853 strains were used as internal controls.

### 2.7. Adherence and Invasion Assays

The N1E-115 (ATCC^®^CRL-2263) cell line was cultured in Dulbecco’s Modified Eagle Medium (DMEM) high glucose 4.5 g/L (ATCC; Manassas, VA, USA), supplemented with 10% fetal bovine serum (FBS) from (ATCC; Manassas, VA, USA). Subsequently, the cells were induced to differentiate in DMEM medium supplemented with 2% FBS and 1.25% dimethyl sulfoxide over a period of five days. Cell cultures were established in 24-well plates (Corning Life Sciences, Tewksbury, USA) at a concentration of 1 × 10^5^ cells/mL and subjected to infection with each *C. sakazakii* isolate at a multiplicity of infection ratio of 100:1 following prior cultivation in Luria broth. Infection was conducted for a duration of 4 h at 37 °C in an atmosphere containing 5% CO_2_. After the incubation period, the cells were rinsed with 1× phosphate-buffered saline (PBS), and the removal of *C. sakazakii* was achieved by introducing 1 mL of 0.1% Triton X-100 (Amresco, Solon, OH, USA). To determine the colony-forming units (CFU) of bacteria attached to the N1E-115 cells, various dilutions were carried out in Luria broth [14]. For the invasion assay, the preparation of the N1E-115 monolayers and the time for infection were as described for the adhesion assay. After 4 h of incubation, the infected monolayers were washed with 1× PBS and incubated with 1 mL of DMEM plus 300 µg/mL lysozyme (Sigma-Aldrich, Virginia Beach, USA) and 100 µg/mL gentamicin (Sigma-Aldrich, USA) for 2 h at 37 °C in 5% CO_2_. The cells were washed three times with 1× PBS, separated with 1 mL of 0.1% Triton X-100, and plated on Luria–Bertani agar. Invasion frequencies were calculated as the number of bacteria that survived incubation with gentamicin and lysozyme divided by the total number of bacteria present in the absence of this antibiotic (bacterial adherence) [18]. Both assays (adhesion, invasion) were repeated twice and performed in duplicate. The data were expressed as the means. The *Cronobacter* strain ATCC BAA-894 was used as the control.

### 2.8. Detection of Antibiotic Resistance and Virulence Genes

The presence of antibiotic resistance genes was determined using the ResFinder tool from the Center for Genomic Epidemiology (CGE) (http://www.genomicepidemiology.org, accessed 28 September 23) [40]. Thresholds for target scanning were set with a required identity of ≥90% to the reference sequence and an aligned reference sequence ≥ 99%. For antimicrobial resistance genes, the Comprehensive Antibiotic Resistance Database (CARD) with the “perfect” and “strict” default settings [41], as well as the Task Template AMRFinderPlus 3.11.2 available in the Ridom SeqSphere+ v.9.0.8 software using the EXACT method at 100% and BLAST alignment for protein identification, were used [42]. 

### 2.9. Detection of Plasmids and Mobile Genetic Elements (MGEs)

The detection of plasmids was carried out using the MOB-suite tool v3.1.4 [43] integrated in Ridom SeqSphere v9.0.8, and the CGE MobileElementFinder v1.0.5 [44] (accessed on 10 September 2023). 

## 3. Results

### 3.1. Sampling and Identification of Isolates

Out of the one-hundred samples subjected to analysis, five presumptive strains of *Cronobacter* spp. were isolated from separate samples. Subsequently, four tested positive for *Cronobacter* spp. using Matrix-Assisted Laser Desorption/Ionization–Time of Flight Mass Spectrometry (MALDI–TOF MS), and one was identified as *Franconibacter helveticus*. Further characterization through ribosomal multilocus sequence typing (rMLST), involving 53 genes, and whole-genome sequencing (WGS) data revealed three isolates as *Cronobacter sakazakii*, one as *Cronobacter malonaticus*, and the remaining isolate was confirmed as *Franconibacter helveticus* (Table 1).

Two strains of *C. sakazakii* ST1 (CC1) and one ST31, CC31 (serotypes *Csak* O:1 and O:2, respectively), and one strain of *C. malonaticus* ST60, CC60 (*Cmal* O:1) were identified by average nucleotide identity and cgMLST. Using the cgMLST scheme, it is observed that, between both ST1 strains, there were only four allele differences (Figure 1).

### 3.2. Antibiotic Resistance Profile

All strains of *C. sakazakii* and *C. malonaticus* were 100% resistant to cefalotin, 75% to ampicillin, 50% to amikacin and ceftazidime, and one strain was resistant to amoxicillin-clavulanic acid (Table 2). *C. sakazakii* ST1 strains exhibited multidrug resistance to four antibiotics (AK, AM, CAZ, and KF), while *C. malonaticus* ST60 strains displayed resistance to three antibiotics (AM, AMC, and KF). The strain of *F. helveticus* only showed resistance to ceftazidime.

### 3.3. Adherence and Invasion Assays 

In our study, 100% of the strains of *C. sakazakii* and *C. malonaticus* adhered to the N1E-115 cell line, with ranges from 1.4 to 8.7 × 10^6^ CFU/mL. Strain 510178-22 (ST31) exhibited the highest level of adherence, while the strain 510177-22 (ST60) showed the lowest. In the invasion assay, only two strains, 510178-22 and 510177-22, invaded with frequencies ranging from 0.0002 to 0.0007%. The control strain ATCC BAA-894 achieved 22 × 10^6^ CFU/mL and 0.15% for adherence and invasion, respectively.

### 3.4. Detection of Antibiotic Resistance and Virulence Genes

A total of 18 resistance genes were detected in *C. sakazakii* strains, and 17 in *C. malonaticus* strains. All *C. sakazakii* strains had bla_CSA-1_, while *C. malonaticus* exhibited bla_CMA-1_, both of which confer resistance to cephalosporins. Only *C. sakazakii* ST1 possessed *mcr-9.1* genes, which confer resistance to colistin. All *C. sakazakii* and *C. malonaticus* strains shared the same efflux genes (*adeF*, *H-NS*, *msbA*, *marA*, *kpnFE*, *emrRB*, *qacG*, *rsmA*, and *CRP*), the antibiotic inactivation gene (*fosA8*), and five antibiotic target alteration genes (*pBP3*, *glpT*, *eF-Tu*, *vanG*, and *AcrAB-TolC*). The *F. helveticus* strain exhibited 13 resistance genes, with *fosA5*, *qacJ*, *marA*, and *AcrAB-TolC* being notable among them (Table 3).

*C. sakazakii* isolates encoded for 34 virulence genes that were detected by WGS and clustered as flagellar proteins, outer membrane proteins, chemotaxis, hemolysins, invasion, plasminogen activator (*cpa*), colonization, transcriptional regulator, survival in macrophages, utilization of sialic acid (*nanA,K,T*), desiccation tolerance (*cheB*, *wzzB*), and toxin–antitoxin genes (*fic*, *relB*). In the *C. malonaticus* strain, 30 virulence-related genes similar to *C. sakazakii* were detected, except for the *cpa* and *nanAKT* cassette genes (Table 4).

### 3.5. Detection of Plasmids and Mobile Genetic Elements (MGEs)

The pESA3 plasmids and seven mobile genetic elements (MGEs) (IS5075, ISEsa2, ISEsa1, IS26, IS903, ISP-pu12, and IS102) were detected in *C. sakazakii* ST1. In the case of *C. sakazakii* ST31, the plasmid pSP291-1, the plasmid pCMA2, and one MGE (ISEsa1) were found. *C. malonaticus* strains harbored the pCMA1 plasmid and one MGE (ISSen4). *F. helveticus* harbored two plasmids, p24.751 and p14.9 with one MGE (ISPpu12) (Table 5 and Figure 2).

## 4. Discussion

The presence of *C. sakazakii* and *C. malonaticus* has been detected in a variety of foods, including infant formula and powdered dairy products intended for children under 2 years of age. However, there is a significant lack of information regarding the contamination of these pathogens in powdered dairy products intended for adult and older adult populations. 

In our study, we found that the prevalence of *Cronobacter* spp. was 4%, and we isolated two strains of *C. sakazakii* ST1 (*Csak* O:1), one strain of *C. sakazakii* ST31 (*Csak* O:2), and one strain of *C. malonaticus* ST60 (*Cmal* O:1). These pathogenic strains have been associated with clinical cases in young and older adults by various authors [9,12,19,45].

CgMLST analysis allowed us to determine that the two isolates of *C. sakazakii* ST1 were closely related, with only a four-allele difference. In a previous study, the same cgMLST scheme enabled us to establish a genotypic relationship between three strains of *C. sa-kazakii* ST1 isolated in 2021 and four ST1 strains of *C. sakazakii* found in a food alert in Chile in 2017, with differences ranging from one to three alleles [46]. In another context, in 2019, as part of a multicenter European study that employed the cgMLST typing technique, eight *C. sakazakii* ST1 isolates were identified. Among these, two isolates from an outbreak in Austria in 2009, which affected two newborns with necrotizing enterocolitis, differed by only one allele. Furthermore, three ST1 isolates in Austria and one in Denmark differed by 203 alleles from the ATCC BAA-894 strain, isolated from an infant formula associated with a fatal case in the United States in 2001 [47]. The use of the standardized *Cronobacter* spp. cgMLST scheme allows for the establishment of genetic relationships even among strains that may not appear related initially. This tool has become essential in outbreak and case investigations, similar to how it is currently used for *Listeria monocytogenes* strains [48].

The overuse of antibiotics in the food chain has become a serious public health issue, leading to the implementation of global prevention campaigns [49]. Therefore, the presence of antibiotic-resistant microorganisms in dairy products consumed by older adults poses a significant risk that requires thorough study and documentation, especially due to the increasing immunological vulnerability associated with aging. In our study, we observed that all strains of *C. sakazakii* and the one of *C. malonaticus* were resistant to cephalothin, an intrinsic resistance that has been widely reported in various previous studies [50,51,52]. Additionally, the two strains of *C. sakazakii* ST1 exhibited multidrug resistance (MDR) to four antibiotics (AK, AM, CAZ, and KF), while the *C. malonaticus* strain was resistant to three antibiotics (AM, AMC, and KF). Bacteria are classified as MDR when they present resistance to three or more families of antibiotics to which they are usually sensitive, such as beta-lactams (penicillins and cephalosporins), carbapenems, aminoglycosides, and quinolones [53,54], as in this study. The presence of MDR strains has become increasingly concerning; for example, several cases of neonatal infection caused by MDR *Cronobacter* spp. strains have been documented, resulting in deaths or severe consequences [55,56]. In other similar studies with powdered dairy products intended for infants, it was found that 96% of *C. sakazakii* strains were MDR, primarily showing resistance to amoxicillin-clavulanic acid, amoxicillin, ampicillin, cefoxitin, cefepime, erythromycin, and ceftriaxone, while being susceptible to trimethoprim/sulfamethoxazole and levofloxacin [57]. In China, it was discovered that *C. sakazakii* strains isolated from powdered infant formulas and processing environments were also resistant to amoxicillin-clavulanic acid, ampicillin, and cefazolin [58], and, in a more recent study, they exhibited resistance to clarithromycin, ampicillin, and trimethoprim/sulfamethoxazole [59].

One of the crucial initial steps in the bacterial pathogenesis process is cellular adhesion [60]. In our study, all strains of *C. sakazakii* and *C. malonaticus* demonstrated the ability to adhere to the N1E-115 ATCC CRL-2263 cell line. Similar results have been observed in previous studies, where it was found that strains of *C. sakazakii* isolated from clinical cases exhibited higher adherence rates (0.915%) compared to strains from other sources of the pathogen (0.0002%) [19]. These values are consistent with what we obtained in our study. However, in terms of the cellular invasion assay, only the strains of *C. sakazakii* ST31 and *C. malonaticus* isolated in our study showed the capacity for cellular invasion, with extremely low invasion frequencies ranging from 0.0002% to 0.0007%. These results contrast significantly with previous reports [61,62,63,64], where higher invasion rates were recorded.

Genomic analysis revealed the presence of several antibiotic resistance genes, primarily grouped into the categories of antibiotic target alteration and antibiotic efflux. All strains of *Cronobacter* spp. examined showed resistance genes for beta-lactamases *CSA-1* and *CMA-1*, which was consistent with the observed in vitro resistance to the cephalosporin KF. Since the initial report on the *bla_CSA_* and *bla_CMA_* gene family responsible for cephalosporin resistance in *Cronobacter* spp., resistance to KF has been documented in various clinical and environmental strains in different countries worldwide [65,66,67,68,69]. Additionally, we detected the presence of the *mrc 9.1* gene, which confers resistance to colistin (polymyxin), considered a gene of significant concern in public health. This gene has been reported in clinical cases associated with *Cronobacter sakazakii* ST1, as well as isolates from foods like powdered infant formula (PIF) [70,71].

The isolates of *C. sakazakii* and *C. malonaticus* shared 30 similar virulence genes identified through whole-genome sequencing (WGS), which were grouped into various categories, such as flagellar proteins, outer membrane proteins, chemotaxis, hemolysins, invasion, colonization, transcriptional regulation, survival in macrophages, desiccation tolerance, and toxin–antitoxin genes. In the *C. malonaticus* strain, the *cpa* gene, which is specifically associated with the species *C. sakazakii* and is related to serum resistance and invasion, was not found [72]. Additionally, the *nanAKT* gene cassette, which encodes exogenous sialic acid utilization, was absent [73]. Sialic acid, a nine-carbon monosaccharide, is an essential component of glycoconjugates and glycoproteins found in various mammalian tissues and plays a crucial role in numerous physiological processes, including brain development and host–pathogen interactions [4]. It also regulates the expression of key enzymes, such as sialidase and adhesins, and its ability to inhibit transcription factors involved in the *fimB* gene, which mediates adhesion and invasion of epithelial cells [74,75].

Flagellum synthesis is crucial for motility, adhesion, and invasion in bacteria, and the detection of flagellar proteins such as FlgE, FlgL, and FliC in the studied strains of *C. sakazakii* and *C. malonaticus* plays a crucial role in their pathogenesis [22,76], as well as the presence of outer membrane proteins like OmpA and OmpX [77,78]. Additionally, we identified the presence of the *fic* gene, which encodes a toxin, and the *relB* gene, responsible for the production of the antitoxin *relE*, both forming part of a bicistronic toxin–antitoxin (TA) operon. These TA systems are small genetic elements found in plasmids, phage genomes, and the chromosomes of various bacterial species. TA genes play a fundamental role in the physiology of bacterial stress response, contributing to the stabilization of horizontally acquired mobile genetic elements and participating in persistence phenotypes in some species, including *E. coli* and *Salmonella* [79]. It has been reported that *fic* and *hipA* follow specific evolutionary patterns in *C. sakazakii*, highlighting that *C. sakazakii* ST1 strains were the only ones containing TA homologs [80].

The two strains of *C. sakazakii* ST1 that we evaluated carried plasmids homologous to pESA3, which have been previously reported due to their association with clinical cases caused by *C. sakazakii* ST1 [81]. Franco et al. [16] reported that plasmid pESA3 encodes a replication origin gene similar to RepFIB (incompatibility class) both unique and shared (*repA*), as well as virulence genes for iron acquisition, a siderophore aerobactin, a type VI secretion system, and the *cpa*-producing gene, a protease capable of degrading host serum, found only in *C. sakazakii* strains. Additionally, we detected the presence of mobile genetic elements (MGEs) such as IS26, which acts as an adapter mediating recombination and transposition events, playing a key role in the formation of large multidrug-resistant (MDR) regions, as found in these ST1 strains [63]. On the other hand, the *C. sakazakii* ST31 strain harbored the pSP291-1 plasmid, which had been previously isolated from the *C. sakazakii* SP291 (ST4) strain. The latter strain is known for its persistent thermotolerance, harboring various heavy metal resistance genes, disinfectants, stress, *cpa* virulence gene, and it was isolated from a powdered infant formula factory [82]. Regarding the *C. malonaticus* ST60 strain, we found the pCMA1 plasmid, which was first documented in 2016. This plasmid is homologous to another one found in the *C. malonaticus* ST7 strain (LMG23826^T^), which was isolated from a clinical case in 1977 [83].

Prior to the era of taxonomy based on whole-genome sequencing, *Franconibacter helveticus* was named as *Enterobacter helveticus*, before being placed temporarily in the *Cronobacter* genus as *Cronobacter helveticus*. It was reclassified in 2014 as *F. helveticus* [3]. The misidentification as *Cronobacter* can not only affect epidemiological statistics but also generate false positives when analyzing powdered infant formula or other dairy products [84].

*C. sakazakii* exhibits a notable capacity to adapt to extremely dry environments, such as powdered dairy food production facilities. Despite the lack of water, these microorganisms have developed strategies to cope with such unfavorable conditions [85]. Therefore, gathering information on the presence of microorganisms like *C. sakazakii and C. malonaticus* in dairy products provides robust evidence to understand the severity of the problem and establish strategies aimed at improving food safety. The implementation of Hazard Analysis and Critical Control Points (HACCP) systems, along with preventive measures in the industry, such as regular monitoring of dry products, rigorous cleaning and disinfection protocols, and the use of packaging and storage techniques that minimize moisture, are essential to mitigate the risk of contamination by *C. sakazakii* and *C. malonaticus* [86]. Further research directions could focus on understanding the molecular mechanisms underlying the adaptability of *C. sakazakii* and *C. malonaticus* to dry conditions and developing more effective control strategies to ensure the safety of dairy products.

## 5. Conclusions

The isolates of *C. sakazakii* and *C. malonaticus* analyzed in this study exhibited high resistance to multiple antibiotics, carrying genes encoding various proteins related to antibiotic resistance, as well as a variety of virulence factors. As a result, powdered dairy products intended for consumption by immunocompromised adults pose a significant health risk. Therefore, strict adherence to good manufacturing practices, good hygiene practices, and the implementation of HACCP in the production of powdered dairy products, along with epidemiological surveillance by health authorities, form the basis of measures to mitigate the risks of future infections.

## Figures and Tables

**Figure 1 microorganisms-11-02841-f001:**
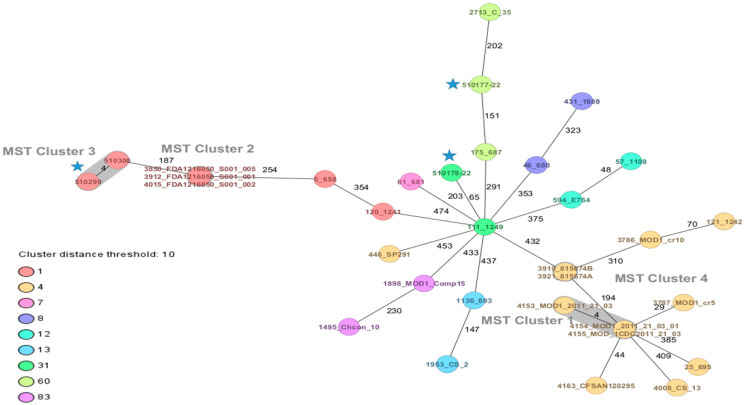
Minimum spanning tree (MST) of three strains of *Cronobacter sakazakii* and one of *Cronobacter malonaticus* from dairy products, complemented with strains of *C. sakazakii* ST1, ST4, ST12, ST13, ST31, ST83 and *C. malonaticus* ST7, ST60 of clinical and food origin. The isolates are represented as colored circles according to their sequence type (ST) as defined using the 7-loci MLST scheme (STs). Black numbers on the connection lines indicate the number of allelic differences between isolates from the *C. sakazakii/malonaticus* cgMLST scheme comprising 2831 target genes. Isolates falling under the cluster threshold of 10 alleles are marked in grey as clusters. Strains of this study 
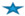
.

**Figure 2 microorganisms-11-02841-f002:**
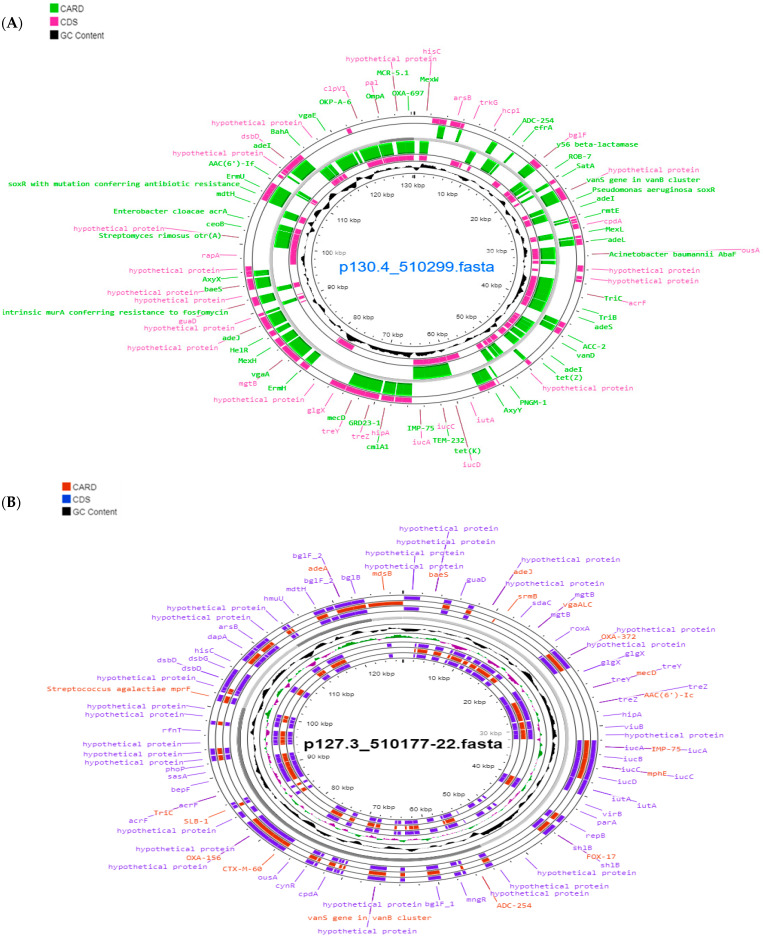
Reconstruction of plasmid p130.4_510299 (pESA3) belonging to the strain *C. sakazakii* 510299 ST1 (**A**) and plasmid p127.3_510177 (pCMA1) of *Cronobacter malonaticus* 510177-22 ST60 (**B**) visualized using Proksee [32].

**Table 1 microorganisms-11-02841-t001:** Identification of *Cronobacter* spp. and *F. helveticus* strains isolated from powdered dairy products by MALDI–TOF MS and rMLST whole-genome sequencing.

Sample ID	PubMLST ID *	MALDI-TOF	WGS rMLST Result **	ST	CC	Serotype (*gnd-galF* Alleles)	Collection Date
510299	4183	*Cronobacter* spp.	*Cronobacter sakazakii*	1	1	*Csak* O:1	2018
510300	4184	*Cronobacter* spp.	*Cronobacter sakazakii*	1	1	*Csak* O:1	2018
510178-22	4187	*Cronobacter* spp.	*Cronobacter sakazakii*	31	31	*Csak* O:2	2019
510177-22	4188	*Cronobacter* spp.	*Cronobacter malonaticus*	60	60	*Cmal* O:1	2019
510441-19	4189	*Franconibacter helveticus*	*Franconibacter helveticus*	345	ND	*Fhelv* O:1	2019

* MLST database ID; ** ribosomal multilocus sequence typing PubMLST database; ST: sequence type; CC: clonal complex; ND: not determined.

**Table 2 microorganisms-11-02841-t002:** Antibiotic resistance profiles of *Cronobacter* spp. strains and *F. helveticus*.

Strains	Species	AK (30 µg)	AM (10 µg)	AMC (20/10 µg)	CAZ (30 µg)	CIP (5 µg)	CL (30 µg)	CTX (30 µg)	GE (10 µg)	KF (30 µg)	TE (30 µg)
510299	*C. sakazakii* ST1	**R**	**R**	S	**R**	S	S	S	S	**R**	S
510300	*C. sakazakii* ST1	**R**	**R**	S	**R**	S	S	S	S	**R**	S
510178-22	*C. sakazakii* ST31	S	S	S	S	S	S	S	S	**R**	S
510177-22	*C. malonaticus* ST60	S	**R**	**R**	S	S	S	S	S	**R**	S
510441-19	*F. helveticus* ST345	S	S	S	**R**	S	S	S	S	S	S

AK: amikacin; AM: ampicillin; AMC: amoxicillin-clavulanic acid; CAZ: ceftazidime; CIP: ciprofloxacin; CL: chloramphenicol; CTX: cefotaxime; GE: gentamicin KF: cephalothin; TE: tetracycline; ST: sequence type; **R**: resistance; S: susceptibility.

**Table 3 microorganisms-11-02841-t003:** Antibiotic resistance gene profiles in the strains under study.

Strain ID	ST	Resistance Profile	Resistance Genes
510299	ST1	AK, AM, CAZ, KF	*mcr-9.1*, *CSA-1*, *adeF*, *CRP*, *emrBR*, *EF-Tu*, *GlpT*, *fosA8*, *H-NS*, *PBP3*, *KpnF*, *marA*, *msbA*, *qacG*, *rsmA*, *AcrAB-TolC with MarR mutations*
510300	ST1	AK, AM, CAZ, KF	*mcr-9.1*, *CSA-1*, *adeF*, *CRP*, *emrB*, *EF-Tu*, *GlpT*, *fosA8*, *H-NS*, *PBP3*, *KpnFE*, *marA*, *msbA*, *qacG*, *rsmA*, *vanG*, *AcrAB-TolC with MarR mutations*
510178-22	ST31	KF	*CSA-1*, *KpnEF*, *marA*, *qacG*, *CRP*, *adeF*, *vanG*, *emrRB*, *rsmA*, *H-NS*, *fosA8*, *msbA*, *PBP3*, *GlpT*, *AcrAB-TolC with MarR mutations*
510177-22	ST60	AM, AMC, KF	*CMA-1*, *adeF*, *rsmA*, *H-NS*, *adeF*, *KpnEF*, *marA*, *vanG*, *CRP*, *qacG*, *fosA8*, *msbA*, *emrBR*, *PBP3*, *GlpT*
510441-19	ND	CAZ	*fosA5*, *qacJ*, *marA*, *KpnFEH*, *adeF*, *baeR*, *rsmA*, *emrR*, *msbA*, *CRP*, *GlpT*, *PBP3*, *AcrAB-TolC with MarR mutations*

*mcr-9.1*: peptide antibiotics; *CSA-1* and *CMA-1*: cephalosporin; *adeF*: fluoroquinolone antibiotic, tetracycline antibiotic; *CRP*: macrolide antibiotic, fluoroquinolone antibiotic, penam; *emrBR:* macrolide antibiotic, fluoroquinolone antibiotic, penam; *EF-TU*: *elfamycin*; *GLPT*: *phosphonic acid*; *fosA:* fosfomicyn; H-NS: macrolide antibiotic, fluoroquinolone antibiotic, cephalosporin, cephamycin, penam, tetracycline; *PBP3*: cephalosporin, cephamycin, penam; *kpnEF*: macrolide, aminoglycoside, cephalosporin, tetracycline, rifamycin, disinfecting agents, and antiseptics; *marA*: fluoroquinolone, monobactam, carbapenem, cephalosporin, glycylcycline, cephamycin, penam, tetracycline, rifamycin, phenicol, penem, disinfecting agents, and antiseptics; *msbA*: nitroimidazole antibiotic; *qacJ*: disinfecting agents and antiseptics; *rsmA*: fluoroquinolone antibiotic, diaminopyrimidine antibiotic, phenicol antibiotic: *AcrAB-TolC*: fluoroquinolone, cephalosporin, glycylcycline, penam, tetracycline, rifamycin, phenicol, disinfecting agents, and antiseptics.

**Table 4 microorganisms-11-02841-t004:** Putative virulence and distribution of other genes in the *C. sakazakii* and *C. malonaticus* strains by whole-genome sequencing.

Virulence Gene	Function	*C. sakazakii* ST1 (510299)	*C. sakazakii* ST1 (510300)	*C. sakazakii* S31 (510178-22)	*C. malonaticus* ST60 (510177-22)
*flgB*	motility	+	+	+	+
*flgK*	flagellar hook-associated protein 1	+	+	+	+
*flgL*	flagellar hook-associated protein 3	+	+	+	+
*flgM*	negative regulator of flagellin synthesis	+	+	+	+
*flgN*	flagellar synthesis FlgN protein	+	+	+	+
*flhD*	flagellar hook-associated protein 2	+	+	+	+
*fliA*	flagellar operon FliA	+	+	+	+
*fliC*	flagellin	+	+	+	+
*fliD*	flagellar hook-associated protein 2	+	+	+	+
*fliR*	flagellar biosynthetic FliR protein	+	+	+	+
*fliT*	flagellar FliT protein	+	+	+	+
*fliZ*	FliZ protein	+	+	+	+
*lolA*	outer membrane lipoprotein carrier protein	+	+	+	+
*motB*	chemotaxis MotA protein	+	+	+	+
*sdiA*	LuxR family transcriptional regulator	+	+	+	+
*slyB*	outer membrane lipoprotein SlyB	+	+	+	+
*tolC*	outer membrane channel protein	+	+	+	+
*msbA*	survival in macrophage	+	+	+	+
*mviN*	protective immunity and colonization	+	+	+	+
*cpa*	plasminogen activator	+	+	+	-
*hem*	hemolysins	+	+	+	+
*ompA*	adhesion cell; biofilm formation	+	+	+	+
*ompX*	adhesion cell	+	+	+	+
*cheR*	chemotaxis protein methyltransferase	+	+	+	+
*cheY*	response regulator of chemotaxis family	+	+	+	+
*cheB*	desiccation tolerance	+	+	+	+
*lpxA*	epithelial cell invasion and lipid A production	+	+	+	+
*nanA,K,T*	exogenous sialic acid utilization	+	+	+	-
*ibpA*	small heat shock protein	+	+	+	+
*wzzB*	desiccation tolerance	+	+	+	+
*fic*	cell filamentation protein	+	+	+	+
*hsp20*	small shock protein	-	-	-	-
*relB*	RelE antitoxin	+	+	+	+

+: presence; -: absence.

**Table 5 microorganisms-11-02841-t005:** Plasmids and mobile genetic elements of *Cronobacter* spp. and *Franconibacter helveticus* strains.

ID Strain	Specie	ST	Plasmid	Plasmid Accession Number	Size (Kb)	Mobile Genetic Elements
510299	*C. sakazakii*	1	pESA3 BR10-DEC	CP000785 CP035364	130,366 13,688	IS5075, ISEsa2, ISEsa1, IS26, IS903, ISPpu12, IS102
510300	*C. sakazakii*	1	pESA3 pCS36-4CPA	CP000785 KM373703	130,600 3959	IS5075, IS26, ISPpu12, IS102, ISEsa2, ISEsa1
510178-22	*C. sakazakii*	31	p109.3 (pSP291-1) p49.7 (pCMA2)	CP004092 CP013942	109,330 49,749	ISEsa1
510177-22	*C. malonaticus*	60	p127.3 (pCMA1)	CP013941	127,318	ISSen4
510441-19	*F. helveticus*	345	p24.7_510441-19 p14.9_510441-19	CP023876 CP035364	24,653 14,863	ISPpu12

## Data Availability

The *Cronobacter sakazakii*, *Cronobacter malonaticus,* and *Franconibacter helveticus* isolates were submitted to https://pubmlst.org/organisms/cronobacter-spp (accessed on 23 September 2023) and have the ID numbers 4183, 4184, 4187, 4188, and 4189.

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
