# Peer review of "Characterization of Cronobacter sakazakii and Cronobacter malonaticus Strains Isolated from Powdered Dairy Products Intended for Consumption by Adults and Older Adults"

_microorganisms, 2023, doi:10.3390/microorganisms11122841_

Round 1

Reviewer 1 Report

Comments and Suggestions for Authors

Author Response

We thank the referee for his comments. However, the referee did not do a point-by-point review as is usually done. Therefore, we separate the paragraphs for a more appropriate response.

Point 1: First, I have doubts about the title "...dairy products intended for consumption by adults and older adults". The abstract states that powdered dairy products pose a risk to healthy and hypersensitive adults. In conclusion, the authors emphasize that this is a threat to immunosuppressed people. This is too general and variable an approach to assessing a group of vulnerable people. It is necessary to specify which group exactly is meant.

Response Point 1: The title has been simplified to consumption for the adult population. The abstract final sentence referring to ‘hypersensitive adults’ is a concluding statement based on the evaluation of the isolates.

Point 2: We consider healthy adults to be  mmunocompetent, so do single Cronobacter cells pose a threat to them? What is the infectious dose for healthy adults? Is the infectious dose different for  mmunosuppressed people? (Please specify in the discussion).

Response Point 2: The concept of ‘infectious dose’ is no longer used, other than in eductional sectors as it infers there is a definite threshold below which no one gets ill. The infectious doce, did not allow for variation according to state of the bacterial cells (ie. Dehydrated, acidified), nor in the food matrix (ie. liquid passes through the stomach rapidly, fatty foods protect bacteria from stomach acid).

Point 3: It is common knowledge that elderly people who stay in hospitals are at risk of Cronobacter spp.  infection. The diet of such people differs from the conventional one and may be based on special artificial foods or with the use of a feeding tube.

Response Point 3: The manuscript does not consider hospitalised adults with special needs. The purpose was to demonstrate the wider exposure of adults to Cronobacter, whereas readers Will have primarily be aware of the organism in the context of feeding newborn babies.

Point 4: The authors describe the study of samples of dairy products available in supermarkets and pharmacies. Food from supermarkets or even pharmacies is available to the entire society. This group cannot be compared with hospital patients. Moreover, the authors did not specify what type of products were evaluated: powdered milk or other.

Response Point 4: The dairy products have been clarified as requested by the referee.

Point 5: Considering the arguments presented above, the idea of a threat to the health of adults and the elderly presented in the work is exaggerated. It is not possible to eliminate the presence of bacteria from food, even pathogenic ones. Of course, food must be safe for consumers by meeting specific microbiological standards. Other requirements apply to food for special purposes, e.g., for infants.Unfortunately, I do not know the standards for Cronobacter in food. It is worth to mention this data,if available.

Response Point 5: The authors have not proposed the elimination of all bacteria from food. The author refers to specific microbiological standards however such standards are known not to reflect current knowledge on emergent pathogens, viruses and non-culturable organisms. Instead such standards reflect what is considered achievable by industry, and the importance regulators place on the risk posed by foodborne pathogens.

Point 6: The prevalence should be indicated in the abstract because it is low, how to accurately assess the risk to public health? For C. malonaticus only 1%. Moreover, I have doubts about the lack of quantitative results on the occurrence of Cronobacter in the tested samples. What is also important is the number of these bacteria present in food.

Response Point 6: There are the papers by Marcelo Brandao (Brazil) on Cronobacter in non-dairy adult foods. The referee is not aware of the food standards for Cronobacter, and is therefore probably not aware that the standard test method is a presence/absence testing method and not enumeration. This is similar to Salmonella testing. Consequently data on the number of Cronobacter in food is not collected

Point 7: Please provide species identification obtained using WGS in the tables. Table 3 lists the names of thevarious genes without explaining their function. This makes the results difficult to understand.

Response Point 7: The names of the species will be added to the tables. The footnote in Table 3 describes the function of each resistance gene and was summarized for better understanding of readers.

Point 8: Why are species and genus names written either in italics or without italics in the manuscript?Likewise, names of the gene (MCR-9.1 and mcr-9.1).Please provide the definition of MDR and be precise about individual antibiotics or classes of antibiotics in the text of the manuscript.

Response Point 8: The original manuscript submitted to the journal has all genus and species names in italics. We do not know why this change occurs in the version that the journal sends us. We will make all the changes associated with it.

The manuscript is being reviewed to ensure that gene names are italicized.

We added the following definition with 2 references: Bacteria are classified as MDR when they present resistance to three or more families of antibiotics to which they are usually sensitive, such as beta-lactams (penicillins and cephalosporins), carbapenems, aminoglycosides and quinolones [48,49], as in this study.

Reviewer 2 Report

Comments and Suggestions for Authors

The manuscript entitled “Characterization of Cronobacter sakazakii and Cronobacter malonaticus Strains Isolated from Powdered Dairy Products Intended for Consumption by Adults and Older Adults” focuses on the characterization of Cronobacter spp. strains isolated from powdered dairy products intended for adult and older adult consumption. The research uses whole genome sequencing (WGS) to identify genes related to antibiotic resistance and virulence. Tools like the CARD platform, ResFinder, and PlasmidFinder are employed. Disk diffusion was used for susceptibility testing. Five presumptive strains were identified; three were C. sakazakii of the clinical pathovar ST1 variety, one was ST31, and the last was C. malonaticus ST60. Additionally, Franconibacter helveticus ST345 was identified. The C. sakazakii ST1 strains displayed multi-drug resistance.

Major comments:

1.     While the manuscript identifies strains and their antibiotic resistance, it could benefit from a deeper exploration into the pathogenicity of these strains, especially in the context of the adult and older adult populations.

2.     Link the identified strains to any known clinical cases or outbreaks would strengthen the paper's significance.

3.     Beyond the recommendation of using water at 70°C, the manuscript could offer more comprehensive guidelines or control measures for the dairy industry.

4.     Visual representations such as phylogenetic trees, heat maps, or tables summarizing strain characteristics would enhance clarity and comprehension.

5.     The manuscript mentions the use of WGS, but it doesn't specify the sequencing depth. Adequate sequencing depth is crucial to ensure accurate identification and characterization of genomic elements.

6.     Relying solely on WGS can introduce errors. The manuscript should consider validating key findings using other molecular techniques, such as PCR or qPCR, especially for clinically significant or novel findings. The genomic data's accuracy relies heavily on the databases and tools used for annotation. The manuscript should specify which databases (and their versions) were used to annotate the genomic sequences and identify genes of interest.

7.     Single nucleotide polymorphisms (SNPs) can play a crucial role in strains' pathogenicity and antibiotic resistance. A detailed SNP analysis, comparing the identified strains with reference genomes, would provide insights into their genetic variability.

8.     The presence of plasmids, especially those carrying antibiotic-resistance genes, is of significant concern. The manuscript mentions using PlasmidFinder but should delve deeper into identified plasmids' characterization and potential implications.

9.     For a robust genomic analysis, comparing the identified strains' genomes with other known genomes (closely related and diverse) would provide a broader context and help pinpoint unique genomic features.

10.  Beyond identifying genes, a functional analysis to predict the identified strains' metabolic pathways and potential phenotypic traits would be valuable.

Minor comments

   - It would enhance the introduction to provide more context about the risks associated with these bacteria in powdered dairy products.

   - A clear statement on the study's novelty or contribution to the existing body of knowledge would be beneficial.

  - The sampling method described is comprehensive, but additional details on post-sampling storage and processing are recommended to clarify potential contamination risks.

   - Specify any controls or reference strains used during analysis for comparison.

   - Elaborate on any steps taken to avoid cross-contamination during sample processing.

   - Consider using tables or figures to summarize key findings, such as strain characteristics, antibiotic resistance profiles, and virulence factors. This would aid in the reader's understanding.

   -in the discussion section,  elaborate more on how the results fit into the broader context of what's already known.

   - Discuss potential mitigation measures or industry implications based on the identified strains and their properties.

-In the conclusion section, consider elaborating on the broader implications for public health, industry practices, and potential future research directions based on the identified risks.

Author Response

We thank the referee for his comments.

Point 1: While the manuscript identifies strains and their antibiotic resistance, it could benefit from a deeper exploration into the pathogenicity of these strains, especially in the context of the adult and older adult populations.

Response Point 1: In addition, regarding pathogenicity aspects such as tests on a cell line and presence of virulence factors, we add more information regarding this.

Point 2: Link the identified strains to any known clinical cases or outbreaks would strengthen the paper's significance.

Response Point 2: I thought the Chinese student outbreak had been included in the main text and the reference was added.

Point 3: Beyond the recommendation of using water at 70°C, the manuscript could offer more comprehensive guidelines or control measures for the dairy industry.

Response Point 3: The following paragraph was added.

C. sakazakii exhibits a notable capacity to adapt to extremely dry environments, such as powdered dairy food production facilities. Despite the lack of water, these microorgan-isms have developed strategies to cope with such unfavorable conditions [79]. Therefore, gathering information on the presence of microorganisms like C. sakazakii and C. malonaticus in dairy products provides robust evidence to understand the severity of the problem and establish strategies aimed at improving food safety. The implementation of Hazard Analysis and Critical Control Points (HACCP) systems, along with preventive measures in the industry such as regular monitoring of dry products, rigorous cleaning and disin-fection protocols, and the use of packaging and storage techniques that minimize mois-ture, are essential to mitigate the risk of contamination by C. sakazakii and C. malonaticus. Further research directions could focus on understanding the molecular mechanisms un-derlying the adaptability of C. sakazakii and C. malonaticus to dry conditions and develop-ing more effective control strategies to ensure the safety of dairy products.

Point 4: Visual representations such as phylogenetic trees, heat maps, or tables summarizing strain characteristics would enhance clarity and comprehension.

Response Point 4: We do not understand what the reviewer requested, the table has a mimimum spannig tree, tables and figures.

Point 5: The manuscript mentions the use of WGS, but it doesn't specify the sequencing depth. Adequate sequencing depth is crucial to ensure accurate identification and characterization of genomic elements.

Response Point 5: Sequencing quality data as Coverage, no. Of the contigs, N50 were included Supplementary Table 1.

Point 6: Relying solely on WGS can introduce errors. The manuscript should consider validating key findings using other molecular techniques, such as PCR or qPCR, especially for clinically significant or novel findings. The genomic data's accuracy relies heavily on the databases and tools used for annotation. The manuscript should specify which databases (and their versions) were used to annotate the genomic sequences and identify genes of interest.

Response Point 6 : The Q30 score of Illumina sequences is >88%, low quality sequences with a Q20 score <20 are removed in the trimming process. In addition the minimum coverage  of 5x, improving the accuracy significantly. Even the most accurate high fidelity Taq polymerases introduces have an error rate ~ 1.5%! Therefore, we dont see a necessity to use a PCR approach for the validation of a DNA sequence.

It is added that Proksee and Prokka was used to annotate the genomes and the software versions used.

Point 7: Single nucleotide polymorphisms (SNPs) can play a crucial role in strains' pathogenicity and antibiotic resistance. A detailed SNP analysis, comparing the identified strains with reference genomes, would provide insights into their genetic variability.

Response Point 7: Genomes were analyzed using AMRFinder, CARD, VirulenceFinder to detect all known resistances (ncluding point mutations)

Genetic variability is shown as a MST based on cgMLST using a recently published typing scheme. Reference genomes were included in the MST

Bacterial SNP analysis is normally used for epidemiological investigations to investigate a collection of strains within a species/serotype. The objective is to distinquish between closely-related and distantly related strains within a species. The method requires the use of a single reference genome which is closely related to the main cluster of interest. This study was concerned with population diversity and therefore bacterial SNP analysis was not appropirate. Furthermore, the physiological, and virulence traits of reference genomes is often unknown.

Point 8: The presence of plasmids, especially those carrying antibiotic-resistance genes, is of significant concern. The manuscript mentions using PlasmidFinder but should delve deeper into identified plasmids' characterization and potential implications.

Response Point 8: We used MobSuite v.3.1.4 to analyze plasmids and not Plasmid Finder. We fixed this bug in the abstract.

We added the following manuscript:

Franco et al. reported that plasmid pESA3 encodes a replication origin gene similar to RepFIB (incompatibility class) both unique and shared (repA), as well as virulence genes for iron acquisition, a siderophore aerobactin, a type VI secretion system, and the cpa-producing gene, a protease capable of degrading host serum, found only in C. sakazakii strains [Franco et al. 2011]. On the other hand, plasmid pSP291-1 was identified in a highly tolerant C. sakazakii strain isolated from an infant formula production plant, harboring various heavy metal resistance genes, disinfectants, and stress, in addition to the cpa virulence gene [Yan et al. 2013].

Point 9: For a robust genomic analysis, comparing the identified strains' genomes with other known genomes (closely related and diverse) would provide a broader context and help pinpoint unique genomic features.

Response Point 9:  A new analysis was performed that included other 39 genomes from various sources and sequence types. MLST Cronobacter Public databases for molecular typing and microbial genome diversity https://pubmlst.org/organisms/cronobacter-spp was used to obtain information.

Point 10: Beyond identifying genes, a functional analysis to predict the identified strains' metabolic pathways and potential phenotypic traits would be valuable.

Response Point 10: Considering the objective of our study, this request is far from it and therefore, we do not consider it.

Point 11: It would enhance the introduction to provide more context about the risks associated with these bacteria in powdered dairy products.

Response Point 11: More information and references are added.

Point 12:  A clear statement on the study's novelty or contribution to the existing body of knowledge would be beneficial.

Response Point 12: There are no studies regarding the microbial safety of powdered dairy products intended for adult nutrition. Therefore, this is the first study that generates the presence of Cronobacter spp and the risk in vulnerable groups. Patrick et al. and Holy et al., have shown the incidence of Cronobacter in various infections in adults, but there are no studies of the probable sources of contamination.

Point 13: The sampling method described is comprehensive, but additional details on post-sampling storage and processing are recommended to clarify potential contamination risks.

Response Point 13: The samples were stored in their unopened cans according to the manufacturer's instructions until analysis. Subsequently, the cans were opened in a laminar flow hood. The elements to obtain the samples were sterilized at 121°C for 15 minutes and the external surfaces of the cans were sprayed with 70% alcohol. The laboratory is certified according to Chilean standards.

Point 14: Specify any controls or reference strains used during analysis for comparison.

Response Point 14: The missing reference strains are added.

Point 15: Elaborate on any steps taken to avoid cross-contamination during sample processing.

Response Point 15: The work was carried out in a certified laboratory, following the established measures within the pathogen isolation procedures. Additionally, it was conducted in a controlled environment within a laminar flow hood, employing sterilized personal protective equipment to prevent direct contamination. Sterile sampling tools were used (121°C for 15 minutes), and 70% alcohol was applied for surface disinfection.

Point 16: Consider using tables or figures to summarize key findings, such as strain characteristics, antibiotic resistance profiles, and virulence factors. This would aid in the reader's understanding.

Response Point 16: The manuscript has 5 tables and 2 figures that summarize the findings.

 Point 17:   -in the discussion section,  elaborate more on how the results fit into the broader context of what's already known.

- Discuss potential mitigation measures or industry implications based on the identified strains and their properties.

-In the conclusion section, consider elaborating on the broader implications for public health, industry practices, and potential future research directions based on the identified risks.

Response Point 17: C. sakazakii exhibits a notable capacity to adapt to extremely dry environments, such as powdered dairy food production facilities. Despite the lack of water, these microorganisms have developed strategies to cope with such unfavorable conditions. Therefore, gathering information on the presence of microorganisms like C. sakazakii and C. malonaticus in dairy products provides robust evidence to understand the severity of the problem and establish strategies aimed at improving food safety. The implementation of Hazard Analysis and Critical Control Points (HACCP) systems, along with preventive measures in the industry such as regular monitoring of dry products, rigorous cleaning and disinfection protocols, and the use of packaging and storage techniques that minimize moisture, are essential to mitigate the risk of contamination by C. sakazakii. Further research directions could focus on understanding the molecular mechanisms underlying the adaptability of C. sakazakii and C. malonaticus to dry conditions and developing more effective control strategies to ensure the safety of dairy products.

Round 2

Reviewer 2 Report

Comments and Suggestions for Authors

The manuscript can be accepted for publication.